# BSC-Based Digital Transformation Strategy Selection and Sensitivity Analysis

**Mahir Oner** [1,2,*], **Ufuk Cebeci** [2] **and Onur Dogan** [3,4]

1　Department of Industrial Engineering, Munzur University, 62000 Tunceli, Türkiye
2　Department of Industrial Engineering, Istanbul Technical University, 34367 Istanbul, Türkiye; cebeciu@itu.edu.tr
3　Department of Management Information Systems, Izmir Bakircay University, 35655 Izmir, Türkiye; onur.dogan@bakircay.edu.tr or onur.dogan@unipd.it
4　Department of Mathematics, University of Padua, 35122 Padua, Italy
*　Correspondence: mahironer@itu.edu.tr

**Abstract:** In today's digital age, businesses are tasked with adapting to rapidly advancing technology. This transformation is far from simple, with many companies facing difficulties navigating new technological trends. This paper highlights a key segment of a comprehensive strategic model developed to address this challenge. The model integrates various planning and decision-making tools, such as the Balanced Scorecard (BSC), Objectives and Key Results (OKR), SWOT analysis, TOWS, and the Spherical Fuzzy Analytic Hierarchy Process (SFAHP). Integrating these tools in the proposed model provides businesses with a well-rounded pathway to manage digital transformation. The model considers human elements, uncertainty management, needs prioritization, and flexibility, aiming to find the optimal balance between theory and practical applications in real-world business scenarios. This particular study delves into the use of SFAHP, specifically addressing the challenge of effectively selecting the most suitable strategy among various options. This approach not only brings a new perspective to digital transformation but also highlights the importance of choosing the right strategy. This choice is crucial for the overall adaptation of businesses. It shows how carefully applying the SFAHP method is key. Combining this with a successful digital transformation strategy is essential. Together, they provide practical and efficient solutions for businesses in a fast-changing technological environment.

**Keywords:** fuzzy AHP; spherical fuzzy sets; decision making; sensitivity analysis; balanced scorecard

**MSC:** 03B52; 03E72

## 1. Introduction

The digital transformation era is ushering in a revolution in how businesses operate, communicate, and compete. This seismic shift is driven by technological advancements, changing customer expectations, and the need for greater efficiency and agility. Companies should respond with urgency and innovation to successfully navigate this evolving landscape. However, this adaptation process is far from straightforward [1].

One of the primary challenges enterprises face during the digital transformation journey is aligning their existing practices with the new technological paradigms [2]. Often, this struggle arises from a lack of understanding, clarity, or a coherent strategy. Companies may find themselves overwhelmed by the magnitude of change required, leading to resistance and inefficiencies within the organization.

In this complex and multifaceted context, exploring innovative approaches to address these challenges is essential. One such promising approach is integrating strategic planning with lean manufacturing techniques. Lean principles, derived from manufacturing, emphasize the elimination of waste, continuous improvement, and a focus on delivering

value to customers. When applied beyond the shop floor and extended into the digital realm, these principles can help streamline processes, reduce inefficiencies, and enhance the overall agility of the organization [3].

Furthermore, the use of strategic planning and performance evaluation tools such as the Balanced Scorecard (BSC), Objectives and Key Results (OKR), SWOT analysis, and its extension TOWS can play a pivotal role [4]. The Balanced Scorecard provides a comprehensive framework for translating the company's strategic objectives into actionable measures, ensuring alignment across the organization. OKR enable clear goal setting and the transparent tracking of progress, fostering accountability and adaptability. SWOT analysis helps in identifying strengths, weaknesses, opportunities, and threats, which is critical for informed decision making, while TOWS extends this analysis by suggesting strategies to capitalize on strengths and opportunities while mitigating weaknesses and threats [5].

Lean manufacturing techniques offer a robust foundation for adapting to the challenges posed by digital transformation [6]. They are rooted in principles prioritizing efficiency, flexibility, and continuous improvement, essential attributes for thriving in the rapidly evolving technological landscape. These principles align well with the objectives of digital transformation, as they empower organizations to streamline processes, reduce waste, and respond swiftly to changing customer demands and market dynamics [7].

However, the successful integration of lean manufacturing techniques into the context of digital transformation is not a standalone solution. It requires a comprehensive approach, encompassing various facets of decision making and strategic planning. One indispensable process that can significantly enhance the effectiveness of this integration is the Fuzzy Analytic Hierarchy Process (FAHP). FAHP is a decision-making methodology that stands out for its ability to address the inherent human subjectivity, uncertainties, and complexities associated with digital transformation efforts [8].

One of the fuzzy sets' most advanced and recent extensions is the spherical fuzzy sets (SFS) proposed by Kutlu Gundogdu and Kahraman [9]. The concept of SFS involves allowing decision makers to extend other variations of fuzzy sets by establishing a membership function on a spherical surface and independently assigning the function inputs with a larger area. This approach enables the independent assignment of parameters within a broader domain to better capture nuances in decision making [10].

The paper's uniqueness lies in introducing an SFAHP method and demonstrating its use within the BSC-based strategy selection. This SFAHP method empowers decision makers to individually express uncertainties in their decision processes through a linguistic evaluation scale rooted in spherical fuzzy sets. The Spherical Fuzzy Analytic Hierarchy Process (SFAHP) is a decision-making methodology that combines the principles of AHP with the concept of spherical fuzzy sets, allowing for a more robust and realistic representation of uncertainty and ambiguity in complex decision problems [11]. Unlike traditional AHP, which relies on crisp numbers and crisp linguistic assessments, SFAHP accommodates the inherent vagueness in human judgments by using fuzzy numbers and linguistic terms described as fuzzy sets on a spherical domain [12]. This approach is precious when decision makers are unsure or hesitant about their preferences or when dealing with imprecise data. SFAHP offers a more comprehensive and flexible framework for decision making, making it a superior choice when addressing real-world problems characterized by uncertainty, imprecision, and vagueness.

In the evolving landscape of digital transformation, existing models often fall short of comprehensively integrating diverse strategic tools. This paper introduces a novel strategic model that uniquely combines the Balanced Scorecard (BSC), Objectives and Key Results (OKR), SWOT analysis, TOWS matrix, and the Spherical Fuzzy Analytic Hierarchy Process (SFAHP). This integration addresses a significant gap in the literature by providing a multifaceted approach to managing digital transformation. Unlike previous studies, the proposed model leverages the precision of SFAHP, a less commonly applied method in this field, offering new insights and solutions to the challenges faced by businesses

during digital transformation. This innovative approach not only contributes to academic discussion but also proposes practical strategies for businesses navigating the complexities of the digital era.

The paper is structured as follows. Section 2 gives the related work to uncover the gap in the literature. Section 3 explains the technical background of the spherical fuzzy sets because of the technique used in the study. Section 4 presents the methodology to rank and select the suitable strategy and the position of the study in the main projection. Section 5 introduces the case study, and results are given in Section 6. Section 7 discusses the what-if analysis results to expand the experiments. Finally, Section 8 concludes the paper.

## 2. Literature Review

In 1992, David Norton and Robert Kaplan introduced the Balanced Scorecard (BSC) as a component of a strategic project management framework [13]. The Kaplan–Norton BSC model emphasizes four well-rounded perspectives: financial, customer, internal business processes, and learning and growth. This model was developed to address the limitations of traditional project management approaches. BSC has gained widespread acceptance among researchers [14–16] and has been applied across various industries, including food [17], financial services [18], education [19], energy [20], healthcare [21], the sports sector [22], tourism [23], and transportation [24]. Today, the Balanced Scorecard is one of the most prominent and influential performance management systems [25,26].

Selection problems can be applied in the context of the BSC to help organizations choose the most appropriate KPIs and strategic objectives to include in their scorecard. Selection problems can be used in BSC for identifying relevant KPIs [27–30], resource allocation [31–33], balancing perspectives [34–36], and prioritizing strategic objectives [34].

When developing a BSC, organizations often have a wide range of potential KPIs to measure performance in each perspective. A selection problem can be used to determine which KPIs are the most relevant and meaningful for measuring progress toward strategic objectives. Various criteria, such as alignment with the strategy, feasibility of measurement, and impact on overall performance, can be considered in the selection process [27–30]. Birdogan and Abuasad [30] proposed an integrated performance evaluation approach using the Balanced Scorecard-DEMATEL approach. The first stage involves determining performance indicators based on the Balanced Scorecard dimensions, while the second stage prioritizes these dimensions and indicators using DEMATEL. Lin et al. [29] explored the application of the BSC to service performance measurements of medical institutions using the AHP and DEMATEL. Four evaluation dimensions and twenty-two indicators of medical service performance measurements were developed based on the BSC concept.

Another application of selection problems in BSC is related to resource allocation. Once the strategic objectives and associated initiatives are identified, organizations may need to decide how to allocate limited resources, such as budget, manpower, and time, among these initiatives. A selection problem can help optimize resource allocation to maximize the achievement of strategic goals while staying within resource constraints [31–33]. Lyu et al. [33] discussed using integrated approaches such as BSC and Fuzzy TOPSIS for performance evaluation in various domains. Herath et al. [31] presented a mathematical model for allocating limited resources in implementing a BSC strategy to prioritize strategic initiatives and calculate the optimal set of BSC targets.

The BSC is designed to provide a balanced view of an organization's performance from various perspectives. A selection problem can be used to ensure that the selected KPIs and strategic objectives adequately represent each perspective. This helps maintain the balance and comprehensiveness of the scorecard [34–36]. Danesh et al. [35] proposed a novel approach that integrates BSC and a three-stage data envelopment analysis model to select appropriate measures for organizational performance evaluation. The BSCs measures were used as input and output variables in the DEA model, and the efficiency variations in different stages helped determine the most suitable measures for each perspective of the BSC. Stavs et al. [34] prioritized four BSC measures using ANP. They designed a conceptual

framework to evaluate green transport performance and supported the implementation of green transport strategies in industrial companies and supply chains.

Organizations may have multiple strategic objectives within each perspective of the BSC. A selection problem can help prioritize these objectives based on their strategic importance, the potential impact on the organization's success, and the available resources [37,38]. This ensures that the most critical objectives are included in the scorecard and receive the necessary attention and resources. Dodangeh et al. [39] proposed a model that determines the measures and objectives in the BSC by using the consensus of the organization's managers and experts' opinions. It then prioritizes the performances of strategic plans in the BSC using the TOPSIS method, a group decision-making model. Nurcanyo et al. [40] developed the BSC strategy map using the AHP method, with input from faculty leaders through ranking and triangulation methods. Fontes et al. [41] integrated AHP and goal programming. They used AHP to evaluate the relative importance of initiatives based on financial indicators and the goal-programming model to select a set of initiatives that maximize earnings and minimize the capital employed.

These studies assume crisp values for criteria weights and performance ratings. However, real-life strategy selection problems include uncertainty and imprecision [41]. This study improves traditional strategy selection problems in BSC with a recent fuzzy extension of AHP, spherical fuzzy AHP. Strategies were determined with SWOT and TWOS by considering the strengths and opportunities of each strategy. Moreover, SFAHP results were discussed with sensitivity analysis to expand the experiments. As a result of the AHP, this study established a relationship to lean principles for business improvement. None of the previous studies have focused on the strategy selection problem in BSC holistically.

In recent developments within the field, spherical fuzzy sets combined with AHP have been increasingly adopted for diverse selection problems. However, a crucial aspect that is often overlooked in these studies is the integration of sensitivity analysis. For instance, Alossta et al. [42] addressed a location selection problem using an integrated AHP-RAFSI approach but did not delve into the sensitivity analysis aspect, which could have further validated their findings. Similarly, Irfan et al. [43] applied AHP and G-TOPSIS approaches to overcome biomass energy barriers, yet the absence of sensitivity analysis in their methodology left room for potential subjectivity in their results. This trend is further evidenced in studies such as those of Bakır and Atalik [44] on e-service quality in the airline industry using Fuzzy AHP and Fuzzy MARCOS, and Li et al. [45] in their failure analysis of offshore wind turbines. Notably, the inclusion of a sensitivity analysis in our study addresses this gap, enhancing the objectivity and reliability of the decision-making process. Our approach, therefore, not only aligns with the current trajectory of research in this domain but also provides a more comprehensive and robust framework for applying SFAHP in various decision-making contexts.

Building upon this foundation, the integration of the Balanced Scorecard (BSC), SWOT, and TOWS analyses with SFAHP is identified as another layer of innovation in strategic planning and decision-making processes. The BSC, developed by Kaplan and Norton, is recognized for its effectiveness in assessing organizational performance from various perspectives [46], while SWOT and TOWS analyses serve as powerful tools for dissecting an organization's internal and external environments [47]. Furthermore, the benefits of deploying BSC within healthcare organizations have been systematically reviewed, highlighting its applicability across different sectors [48]. Additionally, the SWOT-FAHP-TOWS analysis methodology, as applied by Savari and Amghani [49] for developing adaptation strategies among farmers, showcases the potential of these methodologies when integrated. This approach, through its comprehensive and multifaceted analysis, contributes significantly to enhancing the efficacy and strategic depth of decision-making processes.

A few studies adopted the spherical fuzzy sets in AHP for different selection problems. For example, Otay et al. [50] proposed a single-valued SFAHP-WASPAS method to evaluate three outsourcing manufacturers. Kutlu Gundogu and Kahraman [10] applied SFAHP to decide the best site selection for wind power farms among four alternatives. Both studies

neglected sensitivity analysis and thus included more subjectivity. This paper contributes to the literature by applying the SFAHP method within BSC-based strategy selection and sensitivity analysis to decrease the subjectivity and test the what-if conditions.

### 3. Spherical Fuzzy Sets

Spherical fuzzy sets (SFSs) were introduced by Kutlu and Kahrama [9]. These sets combine concepts from neutrosophic fuzzy sets and Pythagorean fuzzy sets. The main difference lies in how hesitancy is defined. In SFSs, hesitancy degrees are limited to a maximum of 1.

**Definition 1.** *Spherical fuzzy sets [9]—the representation of a spherical fuzzy set $\widetilde{A}_S$ defined on the universe of discourse U is depicted in Equation (1):*

$$\widetilde{A}_S = \left\{ \left\langle u, \left( \mu_{\widetilde{A}_S}(u) \right), \nu_{\widetilde{A}_S}(u), \pi_{\widetilde{A}_S}(u) \right) \right\rangle \mid u \in U \right\} \tag{1}$$

*where* $\mu_{\widetilde{A}_S}(u) : U \to [0,1]$, $\nu_{\widetilde{A}_S}(u) : U \to [0,1]$, $\pi_{\widetilde{A}_S}(u) : U \to [0,1]$, *and* $0 \le \mu^2_{\widetilde{A}_S}(x) + \nu^2_{\widetilde{A}_S}(x) + \pi^2_{\widetilde{A}_S}(x) \le 1 \quad \forall u \in U.$

The membership degree, non-membership degree, and hesitancy degree of element $u$ in relation to the spherical fuzzy set $\widetilde{A}_S$ are denoted as $\mu_{\widetilde{A}_S}$, $\nu_{\widetilde{A}_S}$, and $\pi_{\widetilde{A}_S}$, respectively, for each individual $u$.

**Definition 2.** *The geometric distance between two spherical fuzzy numbers $\mu_{\widetilde{A}_S}$ and $\mu_{\widetilde{B}_S}$, located on the surface of a sphere illustrated in Figure 1, is calculated in Equation (2) [51]:*

$$d\left(\widetilde{A}_S, \widetilde{B}_S\right) = arccos\left\{ 1 - \frac{1}{2}\left( \left(\mu_{\widetilde{A}_S} - \mu_{\widetilde{B}_S}\right)^2 + \left(\nu_{\widetilde{A}_S} - \nu_{\widetilde{B}_S}\right)^2 + \left(\pi_{\widetilde{A}_S} - \pi_{\widetilde{B}_S}\right)^2 \right) \right\} \tag{2}$$

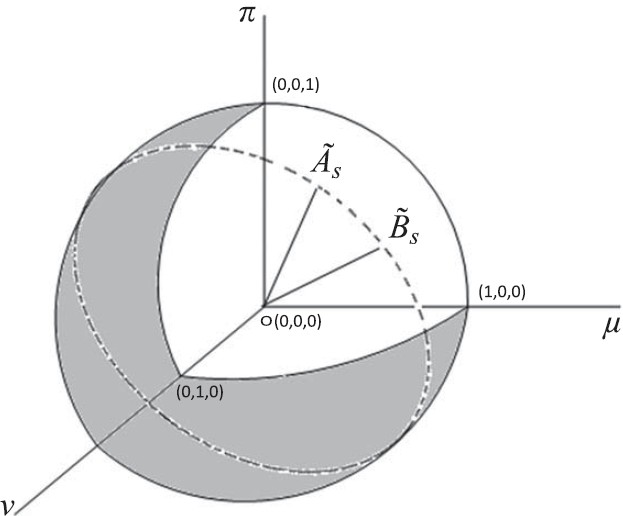

**Figure 1.** Visualization of spherical fuzzy sets in a geometrical context [9].

The transformation of Equation (2) yields the expression for calculating the spherical distance between two spherical fuzzy sets $\widetilde{A}_S$ and $\widetilde{B}_S$ as depicted in Equation (3):

$$d\left(\widetilde{A}_S, \widetilde{B}_S\right) = \frac{2}{\pi}\sum_{i=1}^{n} arccos\left\{ 1 - \frac{1}{2}\left( \left(\mu_{\widetilde{A}_S} - \mu_{\widetilde{B}_S}\right)^2 + \left(\nu_{\widetilde{A}_S} - \nu_{\widetilde{B}_S}\right)^2 + \left(\pi_{\widetilde{A}_S} - \pi_{\widetilde{B}_S}\right)^2 \right) \right\} \tag{3}$$

In order to obtain a distance within the range $[0, 1]$ instead of $\left[0, \frac{\pi}{2}\right]$, the factor $\frac{\pi}{2}$ is applied. The equation for calculating this adjusted distance is presented in Equation (4), which is derived from the equality $\mu_{\widetilde{A}_S}^2 + \nu_{\widetilde{A}_S}^2 + \pi_{\widetilde{A}_S}^2 = 1$:

$$d\left(\widetilde{A}_S, \widetilde{B}_S\right) = \frac{2}{\pi} \sum_{i=1}^{n} arccos\left(\mu_{\widetilde{A}_S}(u_i) \cdot \mu_{\widetilde{B}_S}(u_i) + \nu_{\widetilde{A}_S}(u_i) \cdot \nu_{\widetilde{B}_S}(u_i) + \pi_{\widetilde{A}_S}(u_i) \cdot \pi_{\widetilde{B}_S}(u_i)\right) \qquad (4)$$

The computation of the normalized spherical distance between $\mu_{\widetilde{A}_S}^2$ and $\mu_{\widetilde{B}_S}^2$ on the surface of a sphere involves dividing by the value of $n$, and is expressed in Equation (5):

$$d_n\left(\widetilde{A}_S, \widetilde{B}_S\right) = \frac{2}{n\pi} \sum_{i=1}^{n} arccos\left(\mu_{\widetilde{A}_S}(u_i) \cdot \mu_{\widetilde{B}_S}(u_i) + \nu_{\widetilde{A}_S}(u_i) \cdot \nu_{\widetilde{B}_S}(u_i) + \pi_{\widetilde{A}_S}(u_i) \cdot \pi_{\widetilde{B}_S}(u_i)\right) \qquad (5)$$

It is evident that the value of $d\left(\widetilde{A}_S, \widetilde{B}_S\right)$ lies within the range $0 \leq d\left(\widetilde{A}_S, \widetilde{B}_S\right) \leq n$, and similarly, the value of $d_n\left(\widetilde{A}_S, \widetilde{B}_S\right)$ falls within the interval $0 \leq d_n\left(\widetilde{A}_S, \widetilde{B}_S\right) \leq 1$.

**Definition 3.** *The Spherical Weighted Arithmetic Mean (SWAM) in consideration of the weight vector $w = (w_1, w_2, \cdots, w_n)$ is defined, where each $w_i$ falls within the range $[0, 1]$, and the sum of all $w_i$ is equal to 1 as expressed by the following formulation [10]:*

*Equation (6) outlines the definition of the SWAM operation:*

$$SWAM_w\left(\widetilde{A}_{S1}, \cdots \widetilde{A}_{Sn}\right) = w_1\widetilde{A}_{S1} + w_2\widetilde{A}_{S2} + \cdots + w_n\widetilde{A}_{Sn} =$$
$$\left\{\left[1 - \prod_{i=1}^{n}\left(1 - \mu_{\widetilde{A}_{Si}}^2\right)^{w_i}\right]^{1/2}, \prod_{i=1}^{n}\nu_{\widetilde{A}_{Si}}^{w_i}, \left[\prod_{i=1}^{n}\left(1 - \mu_{\widetilde{A}_{Si}}^2\right)^{w_i} - \prod_{i=1}^{n}\left(1 - \mu_{\widetilde{A}_{Si}}^2 - \pi_{\widetilde{A}_{Si}}^2\right)^{w_i}\right]^{1/2}\right\} \qquad (6)$$

Additional operators, including union, intersection, addition, and multiplication, were introduced in the research by Kutlu and Kahraman [9] and Kutlu and Kahraman [10].

## 4. Methodology

### 4.1. Introducing Main Projection

In this section, the methodology of the study 'BSC-Based Digital Transformation Strategy Selection and Sensitivity Analysis' is detailed. The methodology is anchored in the Balanced Scorecard (BSC) framework, a comprehensive strategic management and performance measurement tool. The BSC framework aids organizations in measuring performance across four key perspectives: financial, customer, internal business processes, and learning and growth [52].

These four perspectives of the BSC enable organizations to define the performance indicators and targets necessary to achieve strategic goals. The framework encompasses not just the financial dimensions of strategic planning but also customer satisfaction, efficiency of internal processes, and corporate learning and growth capabilities. Alongside the BSC framework, our methodology also incorporates SWOT (Strengths, Weaknesses, Opportunities, and Threats) and TOWS (Threats, Opportunities, Weaknesses, and Strengths) analyses. These analyses provide a comprehensive assessment of an organization's internal and external environments, thereby enhancing the effectiveness and comprehensiveness of strategies developed within the BSC framework [53].

Furthermore, in the fourth step of our methodology, the Spherical Fuzzy Analytic Hierarchy Process (SFAHP) is utilized in the strategy determination process. SFAHP addresses the uncertainties and ambiguities in decision-making processes, allowing for a more comprehensive assessment [54]. This approach is particularly effective in complex and multi-criteria decision-making scenarios, commonly preferred in strategy selection and evaluation processes.

This integration represents an innovative approach in both academic and applied strategic management. By combining the strengths of the BSC with SWOT/TOWS analyses and SFAHP, our methodology enables organizations to conduct more comprehensive and

effective strategic planning. This approach enhances the overall effectiveness and reliability of strategic planning, contributing a unique aspect to our study.

The Balanced Scorecard framework integrates the steps given in Figure 2 into a holistic approach to strategic management, providing a well-rounded view of an organization's performance and helping it achieve its long-term vision by translating strategy into actionable plans and measuring results across multiple perspectives.

Vision Definition: This step involves clarifying the organization's long-term vision and strategic objectives, providing all stakeholders with a clear sense of purpose and direction.

SWOT Analysis: SWOT analysis assesses an organization's internal Strengths and Weaknesses as well as external Opportunities and Threats, helping to identify critical factors that impact strategic planning.

TOWS Analysis: An extension of SWOT, TOWS analysis takes the identified strengths, weaknesses, opportunities, and threats and formulates strategies (the TOWS matrix) to capitalize on strengths and opportunities while mitigating weaknesses and threats.

Strategy Definition and Selection: In this phase, strategies are formulated and selected to achieve the organization's goals. The SFAHP can prioritize and choose strategies, considering subjective and uncertain factors.

Perspective Definition: Perspectives are categories or themes representing different aspects of the organization's performance, such as financial, customer, internal processes, and learning and growth. These perspectives provide a structured framework for assessing performance.

Measurement Definition: Key Performance Indicators (KPIs) are defined to measure progress and performance within each perspective quantitatively. These KPIs should align with the selected strategies and vision.

Calculation: Actual performance data are collected and compared to the defined KPIs, often using a scoring system or formula to quantify performance within each perspective.

Creating Action Plans: Based on the performance results, action plans are developed to address areas where performance falls short of targets. These plans outline specific steps and initiatives to improve performance.

Management and Monitoring: Ongoing monitoring and management of performance is crucial. Regular reviews are conducted to track progress, adjust strategies as needed, and ensure alignment with the organization's vision and objectives.

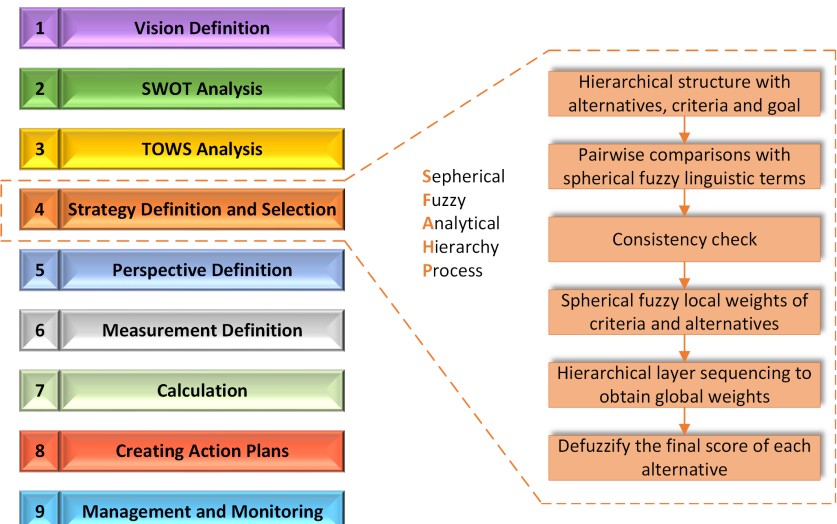

**Figure 2.** Main projection of the strategy selection in the BSC-based digital transformation.

This study focuses on the "Strategy Definition and Selection" step. The defined strategies are ranked using the Spherical Fuzzy AHP method.

*4.2. Spherical Fuzzy AHP (SFAHP)*

*Step 1 Hierarchical structure*

In the initial step, the problem's hierarchy is established. The first level outlines the study's objective, involving the selection of the most appropriate alternative using a scoring index. At the second level, the criteria are defined as $C = \{C_1, C_2, \cdots, C_n\}$, which are utilized to calculate the scoring index. Moving to the third level, there are at least two alternatives $X = \{x_1, x_2, \cdots, x_m\}$, with $m \geq 2$.

*Step 2 Pairwise comparisons with spherical fuzzy linguistic terms*

Constructing pairwise comparison matrices involves employing a spherical fuzzy linguistic evaluation scale for each expert, presented in Table 1 [55]. The scoring indices (SI) are calculated using Equation (7) for the parameters AS, VS, FS, SS, and E, while Equation (8) is used for the parameters SW, FW, VW, and AW [10].

**Table 1.** Spherical fuzzy linguistic evaluation scale [55].

| Linguistic Evaluation | Score Index | $(\mu, v, \pi)$ |
|---|---|---|
| Absolutely Strong (AS) | 9 | (0.9, 0.1, 0) |
| Very Strong (VS) | 7 | (0.8, 0.2, 0.1) |
| Fairly Strong (FS) | 5 | (0.7, 0.3, 0.2) |
| Slightly Strong (SS) | 3 | (0.6, 0.4, 0.3) |
| Exactly Equal (E) | 1 | (0.5, 0.4, 0.4) |
| Slightly Weak (SW) | 1/3 | (0.4, 0.6, 0.3) |
| Fairly Weak (FW) | 1/5 | (0.3, 0.7, 0.2) |
| Very Weak (VW) | 1/7 | (0.2, 0.8, 0.1) |
| Absolutely Strong (AW) | 1/9 | (0.1, 0.9, 0) |

$$SI = \sqrt{\left| 100 \times \left[ \left( \mu_{\tilde{A}_S} - \pi_{\tilde{A}_S} \right)^2 - \left( v_{\tilde{A}_S} - \pi_{\tilde{A}_S} \right)^2 \right] \right|} \tag{7}$$

$$\frac{1}{SI} = \frac{1}{\sqrt{\left| 100 \times \left[ \left( \mu_{\tilde{A}_S} - \pi_{\tilde{A}_S} \right)^2 - \left( v_{\tilde{A}_S} - \pi_{\tilde{A}_S} \right)^2 \right] \right|}} \tag{8}$$

*Step 3 Consistency check*

The evaluation of decision-makers' consistency is conducted by translating the linguistic assessments in a pairwise comparison matrix into scoring indices (SI). Afterwards, the standard Consistency Ratio (CR) is checked against a predefined 10% threshold [56].

Achieving perfect consistency might be challenging, but the goal is to minimize inconsistencies to a level where the decision-making process remains reliable and robust. It is crucial to balance efforts to improve consistency with the practical constraints and complexities of real-world decision scenarios. When consistency was not achieved in the context of SF-AHP, this study chose the "revising judgments" way. This way encourages decision makers to revisit their judgments and pairwise comparisons. Sometimes, inconsistencies stem from errors or oversights in the initial assessments. Re-evaluating and refining these comparisons could enhance consistency. Apart from "revising judgments", some recent developments over the consistency of AHP were presented in [57].

*Step 4 Spherical fuzzy local weights of criteria and alternatives*

The calculations for criteria weights and the weights assigned to each criterion's alternatives are performed using the SWAM operator as described in Equation (6). The spherical fuzzy weights are established utilizing weighted arithmetic means.

*Step 5 Hierarchical layer sequencing to obtain global weights*

The spherical fuzzy weights are consolidated at each level within the hierarchy to establish the final rankings of alternatives, beginning from the lower level (alternatives) and progressing to the upper level (goal). The aggregation of final ranking scores involves

two primary steps. Firstly, criteria weights are converted from fuzzy values using the score function (*S*) outlined in Equation (9). Secondly, the normalization of these weights is achieved utilizing Equation (10). The multiplication operator, as described in Equation (11), is applied in this process, referencing the methodology detailed in [10]:

$$S(\widetilde{w}_j^s) = \sqrt{\left| 100 \times \left[ \left( 3\mu_{\widetilde{A}_S} - \frac{\pi_{\widetilde{A}_S}}{2} \right)^2 - \left( \frac{\nu_{\widetilde{A}_S}}{2} - \pi_{\widetilde{A}_S} \right)^2 \right] \right|} \tag{9}$$

$$\overline{w}_j^s = \frac{S(\widetilde{w}_j^s)}{\sum_{J=1}^n S(\widetilde{w}_j^s)} \tag{10}$$

$$\widetilde{A}_{S_{ij}} = \overline{w}_j^s \cdot \widetilde{A}_{S_i} = \left\langle \left( 1 - \left( 1 - \mu_{\widetilde{A}_S}^2 \right)^{\overline{w}_j^s} \right)^{1/2}, \nu_{\widetilde{A}_S}^{\overline{w}_j^s}, \left( \left( 1 - \mu_{\widetilde{A}_S}^2 \right)^{\overline{w}_j^s} - \left( 1 - \mu_{\widetilde{A}_S}^2 - \pi_{\widetilde{A}_S}^2 \right)^{\overline{w}_j^s} \right)^{1/2} \right\rangle, \forall i \tag{11}$$

To compute the ultimate spherical fuzzy AHP score $\widetilde{F}$, Equation (12) is employed. This process involves utilizing spherical fuzzy arithmetic addition for aggregating the global weights:

$$\widetilde{F} = \sum_{j=1}^n \widetilde{A}_{S_{ij}} = \widetilde{A}_{S_{i1}} \oplus \widetilde{A}_{S_{i2}} \oplus \cdots \oplus \widetilde{A}_{S_{in}}, \ \forall i \tag{12}$$

where

$$\widetilde{A}_{S_{i1}} \oplus \widetilde{A}_{S_{i2}} = \left\langle \begin{array}{c} \left( \mu_{\widetilde{A}_{S_{11}}}^2 + \mu_{\widetilde{A}_{S_{12}}}^2 - \mu_{\widetilde{A}_{S_{11}}}^2 \mu_{\widetilde{A}_{S_{12}}}^2 \right)^{1/2}, \nu_{\widetilde{A}_{S_{11}}}^2 \nu_{\widetilde{A}_{S_{12}}}^2, \\ \left( \left( 1 - \mu_{\widetilde{A}_{S_{12}}}^2 \right) \pi_{\widetilde{A}_{S_{11}}}^2 + \left( 1 - \mu_{\widetilde{A}_{S_{11}}}^2 \right) \pi_{\widetilde{A}_{S_{12}}}^2 - \pi_{\widetilde{A}_{S_{11}}}^2 \pi_{\widetilde{A}_{S_{12}}}^2 \right)^{1/2} \end{array} \right\rangle$$

*Step 6 Defuzzify the final score of each alternative*
The ultimate score for each alternative is subjected to defuzzification by applying the score function as defined in Equation (9). Subsequently, the alternatives are ranked based on the defuzzified scores, where higher values indicate better performance. In the methodology, selecting the finest alternative involves considering both the highest membership degree and the lowest non-membership degree. When alternatives possess equal membership values, a higher hesitancy value is favored over a larger non-membership value.

## 5. Case Study

In this study, the case of a plastic injection manufacturing company was selected to facilitate a better understanding of the paper's focus through a detailed examination of the complexities of digital transformation and strategic decision-making processes. This sector, necessitating rapid adaptation to technological innovations and continuous technical development, presents an ideal context for analyzing digital transformation strategies. The company embodies the challenges and opportunities typical in the digital transformation journey, enabling a more nuanced understanding of strategic decisions via the application of analytical tools. This case is particularly well suited for highlighting the importance of strategic decisions in digital transformation initiatives and contributes to evaluating the effectiveness of various methods.

### 5.1. Data Collection

The Smart Industry Readiness Index (SIRI) Maturity Model was employed to gain a clear understanding of the business's current status. This particular model was chosen because it offers a thorough framework for evaluating the business across three essential dimensions:

- Technology usage;
- Organizational structure;
- Production processes.

In the assessment of the current state of the business within this study, the Smart Industry Readiness Index (SIRI) Maturity Model was utilized, centering on three pivotal dimensions: technology usage, organizational structure, and production processes. The rationale behind focusing on these particular dimensions is firmly rooted in a rich body of literature that underscores their integral role in digital transformation. For instance, the transformative impact of technology on business models and competitive edge is comprehensively explored by Bharadwaj et al. [58], while the pivotal role of organizational structure in the success of the digital transformation is elaborated by O'Reilly and Tushman [59]. Moreover, Porter and Heppelmann's work [60] sheds light on the criticality of digitalization in production processes, particularly in terms of enhancing operational efficiency and effectiveness. Such alignment with these scholarly insights ensures that our analysis is both comprehensive and firmly anchored in established concepts of digital transformation.

### 5.2. Alternatives and Criteria

This section identifies and analyzes alternatives and criteria that guide the decision-making process. Two fundamental frameworks are employed for this purpose: SWOT and TOWS analyses are utilized to identify alternatives, and the Balanced Scorecard is used to determine the criteria.

### 5.2.1. Determination of Alternatives

After the data collection phase, the findings were utilized to conduct a SWOT analysis to identify the company's strengths, weaknesses, opportunities, and threats. The key elements are as follows:

- Strengths: Skilled staff and quality products.
- Weaknesses: Old production tools and low automation.
- Opportunities: Growing market and new technologies.
- Threats: High competition and fluctuating raw material prices.

The selection of these elements for the SWOT analysis was based on a comprehensive evaluation of the company's internal capabilities and external market environment. The strengths and weaknesses were identified through an internal assessment, considering factors such as staff skills and production tools, while the opportunities and threats were determined by analyzing external market trends and competitive dynamics.

The identification of 'skilled staff and quality products' as strengths reflects the company's internal human resource capabilities and product excellence. Conversely, 'old production tools and low automation' were recognized as weaknesses due to their impact on operational efficiency and competitive positioning. 'Growing market and new technologies' were seen as opportunities, highlighting the potential for expansion and technological advancement. Lastly, 'high competition and fluctuating raw material prices' were identified as threats, underscoring the challenges posed by market competition and supply chain volatility.

This analytical approach aligns with the strategic management literature, emphasizing the importance of understanding both the internal and external aspects of a business when formulating strategies. The SWOT analysis thus serves as a foundational step in developing strategic initiatives, ensuring they are well grounded in the realities of the company's operating environment.

Building on the SWOT analysis, Table 2 was developed to transform these elements into actionable strategies.

**Table 2.** Actionable strategies obtained from SWOT and TOWS.

| | **Opportunities** (Growing market, New technologies) | **Threats** (High competition, Fluctuating raw material prices) |
|---|---|---|
| **Strengths** (Skilled staff, Quality products) | Staff Training and Development | Supply chain improvements |
| | Enhance Marketing and Sales | |
| **Weaknesses** (Old production tools, Low automation) | Develop sustainable production practices | Comprehensive technology and automation investment |
| | Improve CRM | |

The strategies identified in the TOWS matrix serve as the alternatives for the next phase, the SFAHP study, to determine their priority ranking. In Table 3, these strategies are further elaborated alphabetically, indicating their origin from the SWOT or TOWS analyses and their intended outcomes.

**Table 3.** Alternatives (strategies) for SFAHP obtained from SWOT and TOWS.

| **Strategies** | **From SWOT** | **From TOWS** | **Outcome** |
|---|---|---|---|
| Advancing Team Skills and Training Initiatives ($A_1$) | Strength: Skilled staff Opportunity: New technologies. | The strategy aims to use skilled staff and new technologies. | Help staff adapt to new technology while improving their existing skills. |
| Boosting Sustainable Production Methods ($A_2$) | Weakness: Old production tools Opportunity: Growing market | The strategy aims to replace or upgrade old production tools to meet the demands of a growing market. | Focus on implementing sustainable production practices to seize opportunities in a growing market. |
| Customer Relationship Enhancement Techniques ($A_3$) | Weakness: Low automation. Opportunity: Growing market | The strategy aims to increase automation and tap into the growing market. | Improve CRM and use automation to grow in the market. |
| Investment in Integrated Technology and Automation ($A_4$) | Weakness: Old tools, Low automation Threat: High competition | The strategy aims to fix old tools and face high competition. | Modernize tools and increase automation to deal with competition. |
| Refining Marketing and Sales Approaches ($A_5$) | Strength: Quality products Opportunity: Growing market | Use product quality to improve marketing and sales strategies. | Aim to gain more market share through high-quality products. |
| Upgrading Supply Network Processes ($A_6$) | Strength: Quality products Threat: Changing raw material prices. | The strategy aims to use quality to deal with changing material costs. | Improve the supply chain to deal with cost changes. |

5.2.2. Determination of Criteria

For determining the criteria for evaluation, this study uses Kaplan and Norton's Balanced Scorecard [61], focusing on four key perspectives: financial, customer, internal process, and learning and growth. These four areas are a basic guide for how the organization is doing. Yet, it is known that every industry has its unique points that need extra care. In plastic injection manufacturing, changes to these guidelines were made based on talks with experts (Expert1 and Expert2). Figure 3 depicts the determined criteria by experts.

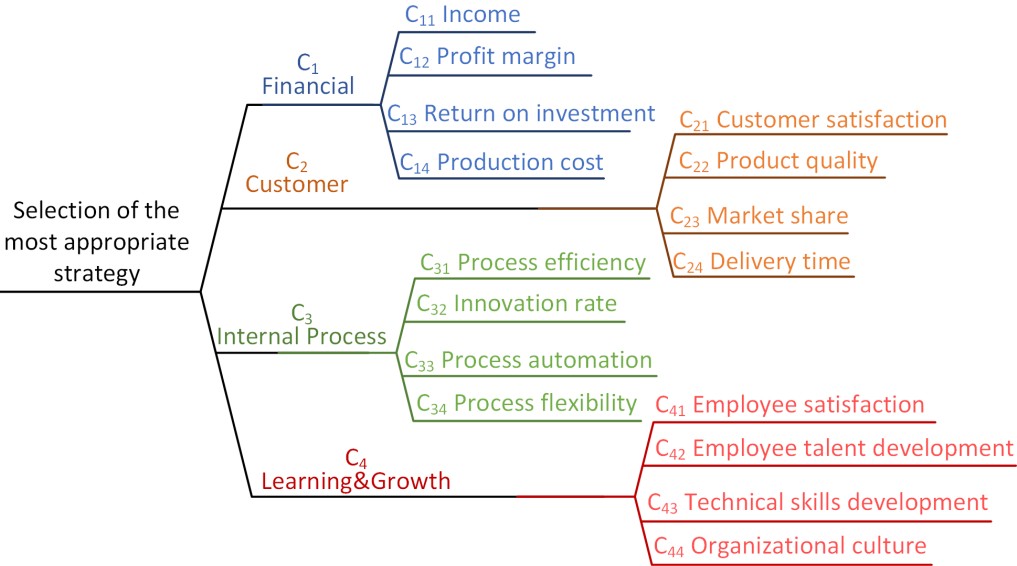

**Figure 3.** Hierarchy of criteria.

- Financial Perspective: The foundational criteria, such as revenue, profit margin, and return on investment (ROI) are retained. Experts have recommended adding "production costs" to underscore its pivotal role in cost management within this sector.
- Customer Perspective: Criteria such as customer satisfaction and market share continue to be recognized. Experts have suggested the inclusion of "product quality" and "delivery time" in the criteria. Product quality is acknowledged as a crucial contributor to customer satisfaction, and timely delivery is seen as a potential competitive edge in the market.
- Internal Process Perspective: The established criteria of process efficiency and innovation rate are kept. Additionally, the "automation of production processes" and "flexibility of production processes" have been proposed by experts. Automation is known to directly affect both efficiency and quality, while production flexibility is crucial for promptly meeting the varying demands of customers.
- Learning and Growth Perspective: Employee satisfaction and employee skill development criteria are sustained. "Technical skill development" has been recommended as a new criterion, reflecting the increasing focus on technological capabilities and automation in the sector. "Organizational culture" has also been introduced as a critical criterion highlighting the influence of company values, norms, and practices on employee performance and development. Cultivating a robust organizational culture is essential for driving technological advancement and preserving a competent workforce.

## 6. Experimental Results

After establishing the hierarchy of the technology selection problem in the fourth step of the main projection in Figure 2, the next step is to construct the pairwise comparison matrices, employing a spherical fuzzy linguistic evaluation scale for each expert, presented in Table 1. Table 4 gives the pairwise comparisons of the superior criteria $C_1$, $C_2$, $C_3$, and $C_4$ for each decision maker. Their linguistic decisions are converted into fuzzy numbers and then fuzzy weights $\widetilde{w}^s$ to understand the importance of the superior criteria if the decisions are consistent ($CR_{DM_1} = 0.059$, $CR_{DM_2} = 0.067$, $CR_{DM_3} = 0.044$). At the end of this step, aggregated and defuzzified weights $\overline{w}^s$ are obtained. They are 0.30, 0.16, 0.31, and 0.23 for $C_1$, $C_2$, $C_3$, and $C_4$, respectively.

**Table 4.** Pairwise comparison of superior criteria.

| | | $C_1$ | $C_2$ | $C_3$ | $C_4$ | $\widetilde{w}^s$ | $\overline{w}^s$ |
|---|---|---|---|---|---|---|---|
| | $C_1$ | E | FS | SW | SW | (0.67, 0.31, 0.25) | 0.32 |
| $DM_1$ | $C_2$ | FW | E | FW | FW | (0.51, 0.47, 0.29) | 0.24 |
| | $C_3$ | SS | FS | E | E | (0.33, 0.65, 0.26) | 0.15 |
| | $C_4$ | SS | FS | E | E | (0.62, 0.37, 0.28) | 0.29 |
| | $C_1$ | E | AS | SS | VS | (0.69, 0.28, 0.26) | 0.34 |
| $DM_2$ | $C_2$ | AW | E | FW | SW | (0.46, 0.53, 0.31) | 0.21 |
| | $C_3$ | SW | FS | E | FS | (0.37, 0.61, 0.29) | 0.17 |
| | $C_4$ | VW | SS | FW | E | (0.59, 0.37, 0.33) | 0.28 |
| | $C_1$ | E | FS | SW | SS | (0.65, 0.34, 0.28) | 0.31 |
| $DM_3$ | $C_2$ | FW | E | AW | SW | (0.55, 0.41, 0.33) | 0.25 |
| | $C_3$ | SS | AS | E | AS | (0.33, 0.65, 0.26) | 0.15 |
| | $C_4$ | SW | SS | AW | E | (0.6, 0.37, 0.3) | 0.28 |
| | | | | | $C_1$ | (0.89, 0.04, 0.32) | 0.30 |
| | Aggregation | | | | $C_2$ | (0.59, 0.24, 0.35) | 0.16 |
| | | | | | $C_3$ | (0.92, 0.03, 0.29) | 0.31 |
| | | | | | $C_4$ | (0.76, 0.11, 0.38) | 0.23 |

$CR_{DM_1} = 0.059$, $CR_{DM_2} = 0.067$, $CR_{DM_3} = 0.044$.

The same procedure is repeated for the sub-criteria to find their importance weights. Table 5 presents a sample illustration of pairwise comparisons of the sub-criteria $C_1$. The weights of $C_{11}$, $C_{12}$, $C_{13}$, and $C_{14}$ were found to be 0.33, 0.23, 0.16, and 0.28, respectively.

**Table 5.** Pairwise comparison of sub-criteria of $C_1$.

| | | $C_{11}$ | $C_{12}$ | $C_{13}$ | $C_{14}$ | $\widetilde{w}^s$ | $\overline{w}^s$ |
|---|---|---|---|---|---|---|---|
| | $C_{11}$ | E | FS | VS | SS | (0.67, 0.31, 0.25) | 0.32 |
| $DM_1$ | $C_{12}$ | FW | E | FS | SW | (0.51, 0.47, 0.29) | 0.24 |
| | $C_{13}$ | VW | FW | E | VW | (0.33, 0.65, 0.26) | 0.15 |
| | $C_{14}$ | SW | SS | VS | E | (0.62, 0.37, 0.28) | 0.29 |
| | $C_{11}$ | E | VS | VS | E | (0.69, 0.28, 0.26) | 0.34 |
| $DM_2$ | $C_{12}$ | VW | E | SS | SW | (0.46, 0.53, 0.31) | 0.21 |
| | $C_{13}$ | VW | SW | E | FW | (0.37, 0.61, 0.29) | 0.17 |
| | $C_{14}$ | E | SS | FS | E | (0.59, 0.37, 0.33) | 0.28 |
| | $C_{11}$ | SS | VS | SS | SS | (0.65, 0.34, 0.28) | 0.31 |
| $DM_3$ | $C_{12}$ | E | FS | E | SW | (0.55, 0.41, 0.33) | 0.25 |
| | $C_{13}$ | FW | E | VW | AS | (0.33, 0.65, 0.26) | 0.15 |
| | $C_{14}$ | E | VS | E | E | (0.6, 0.37, 0.3) | 0.28 |
| | | | | | $C_{11}$ | (0.91, 0.03, 0.3) | 0.33 |
| | Aggregation | | | | $C_{12}$ | (0.77, 0.1, 0.38) | 0.23 |
| | | | | | $C_{13}$ | (0.56, 0.26, 0.35) | 0.16 |
| | | | | | $C_{14}$ | (0.86, 0.05, 0.35) | 0.28 |

$CR_{DM_1} = 0.091$, $CR_{DM_2} = 0.057$, $CR_{DM_3} = 0.045$.

Table 6 indicates the sub- and superior criteria weights. Local weights represent the importance of the sub-criteria, ignoring the superior criteria weights. In contrast, global weights give the overall importance, calculated by the multiplication of the local weights and superior criteria weights.

**Table 6.** Criteria weights.

| | Aggregated Weights | Local Weights | Global Weights | | Aggregated Weights | Local Weights | Global Weights |
|---|---|---|---|---|---|---|---|
| $C_1$ | (0.89, 0.04, 0.32) | - | 0.33 | $C_3$ | (0.92, 0.03, 0.29) | - | 0.31 |
| $C_{11}$ | (0.91, 0.03, 0.3) | 0.33 | 0.1 | $C_{31}$ | (0.77, 0.11, 0.38) | 0.23 | 0.07 |
| $C_{12}$ | (0.77, 0.1, 0.38) | 0.23 | 0.07 | $C_{32}$ | (0.88, 0.05, 0.32) | 0.29 | 0.09 |
| $C_{13}$ | (0.56, 0.26, 0.35) | 0.16 | 0.05 | $C_{33}$ | (0.9, 0.04, 0.31) | 0.29 | 0.09 |
| $C_{14}$ | (0.86, 0.05, 0.35) | 0.28 | 0.09 | $C_{34}$ | (0.66, 0.18, 0.37) | 0.19 | 0.06 |
| $C_2$ | (0.59, 0.24, 0.35) | - | 0.16 | $C_4$ | (0.76, 0.11, 0.38) | - | 0.23 |
| $C_{21}$ | (0.93, 0.03, 0.27) | 0.33 | 0.05 | $C_{41}$ | (0.88, 0.05, 0.33) | 0.29 | 0.06 |
| $C_{22}$ | (0.66, 0.19, 0.36) | 0.19 | 0.03 | $C_{42}$ | (0.83, 0.07, 0.36) | 0.28 | 0.06 |
| $C_{23}$ | (0.89, 0.04, 0.32) | 0.29 | 0.05 | $C_{43}$ | (0.81, 0.08, 0.38) | 0.25 | 0.06 |
| $C_{24}$ | (0.7, 0.16, 0.37) | 0.20 | 0.03 | $C_{44}$ | (0.65, 0.17, 0.39) | 0.19 | 0.04 |

Each alternative is evaluated considering all sub-criteria. Table 7 presents a sample illustration of pairwise comparisons of alternatives with respect to the sub-criteria $C_{11}$. Here, $A_1$'s defuzzified weights $\overline{w}^s$ are 0.16, 0.11, and 0.20 for $DM_1$, $DM_2$, and $DM_3$, respectively. These defuzzified weights are multiplied by 0.33, the weight of sub-criterion $C_{11}$ in Table 5, to obtain the final $\overline{w}^S$.

**Table 7.** Pairwise comparison of alternatives in terms of criterion $C_{11}$.

| | | $A_1$ | $A_2$ | $A_3$ | $A_4$ | $A_5$ | $A_6$ | $\widetilde{w}^s$ | $\overline{w}^s$ | $\overline{w}^S$ |
|---|---|---|---|---|---|---|---|---|---|---|
| | $A_1$ | E | E | VS | AW | E | SS | (0.17, 0.15, 0.19) | 0.16 | 0.05 |
| | $A_2$ | E | E | SW | SS | FW | SS | (0.17, 0.17, 0.21) | 0.15 | 0.05 |
| $DM_1$ | $A_3$ | VW | SS | E | FS | VW | FS | (0.18, 0.16, 0.17) | 0.17 | 0.06 |
| | $A_4$ | AS | SW | FW | E | FS | VS | (0.21, 0.11, 0.17) | 0.21 | 0.07 |
| | $A_5$ | E | FS | VS | FW | E | SW | (0.19, 0.14, 0.18) | 0.18 | 0.06 |
| | $A_6$ | SW | SW | FW | VW | SS | E | (0.14, 0.20, 0.18) | 0.12 | 0.04 |
| | $A_1$ | E | VW | SS | AW | E | VW | (0.12, 0.21, 0.18) | 0.11 | 0.04 |
| | $A_2$ | VS | E | FW | FS | E | SS | (0.20, 0.13, 0.18) | 0.20 | 0.06 |
| $DM_2$ | $A_3$ | SW | FS | E | FW | SW | FW | (0.16, 0.19, 0.18) | 0.15 | 0.05 |
| | $A_4$ | AS | FW | FS | E | FW | FS | (0.20, 0.12, 0.17) | 0.19 | 0.06 |
| | $A_5$ | E | E | SS | FS | E | FW | (0.18, 0.15, 0.22) | 0.17 | 0.06 |
| | $A_6$ | VS | SW | FS | FW | FS | E | (0.19, 0.14, 0.17) | 0.19 | 0.06 |
| | $A_1$ | E | VS | E | VS | SW | SS | (0.21, 0.12, 0.19) | 0.20 | 0.07 |
| | $A_2$ | VW | E | FW | FW | VW | SW | (0.11, 0.23, 0.18) | 0.09 | 0.03 |
| $DM_3$ | $A_3$ | E | FS | E | FS | SS | VS | (0.22, 0.12, 0.19) | 0.22 | 0.07 |
| | $A_4$ | VW | FS | FW | E | VS | FW | (0.16, 0.16, 0.17) | 0.15 | 0.05 |
| | $A_5$ | SS | VS | SW | VW | E | FW | (0.17, 0.17, 0.18) | 0.16 | 0.05 |
| | $A_6$ | SW | SS | VW | FS | FS | E | (0.17, 0.16, 0.19) | 0.17 | 0.06 |

$CR_{DM_1} = 0.018$, $CR_{DM_2} = 0.018$, $CR_{DM_3} = 0.096$.

The subsequent section presents the empirical findings derived from applying the SFAHP to assess the strategic alternatives. Table 8 encapsulates the outcome of this multi-criteria decision-making exercise, delineating the calculated weights, rankings, and relative importance scores of the identified strategic initiatives. Table 8 provides a strategic roadmap for the organization, illustrating the decision-makers' consensus on the relative significance of various initiatives. It is essential to understand that these results reflect the varying levels of importance assigned by the individuals involved, rooted in their personal judgments. To ensure the credibility of these rankings, sensitivity analysis is crucial. This process helps to see how reliable these rankings are by examining the effect of changing opinions or additional information on the prioritization of strategies. The aim is to offer a clearer understanding of the steadiness of these strategies and how well the decision-making process can handle different views.

Advancing Team Skills and Training Initiatives ($A_1$) with a weight of 0.14 and ranking fifth, signifies a moderate priority in the current strategic plan. This reflects the organization's view of the relative importance of team development and training in achieving long-term goals. Boosting Sustainable Production Methods ($A_2$), assigned the lowest weight (0.12) and ranked sixth, indicates a current lesser focus on sustainability within the strategic framework. However, this area may gain more importance in future strategies, reflecting an evolving focus on sustainable practices. Customer Relationship Enhancement Techniques ($A_3$), with a weight of 0.18 and second-place ranking, underscores the strategic focus on customer engagement, highlighting its role as a critical factor for business success. Investment in Integrated Technology and Automation ($A_4$), holding the highest weight of 0.23 and top ranking, demonstrates its central role in the organization's digital transformation journey, emphasizing the critical importance of technology and automation in gaining a competitive advantage and enhancing efficiency. Refining Marketing and Sales Approaches ($A_5$), weighted at 0.16 and ranking fourth, indicates the significant role of marketing and sales activities in the digital transformation process, albeit not as critical as the top-ranked strategies. Upgrading Supply Network Processes ($A_6$), with a weight of 0.17 and third-place ranking, shows the importance of supply chain and logistics processes in the overall strategy, highlighting their role in operational efficiency and customer service.

**Table 8.** Weights and ranking of alternatives.

|  | Weight | Ranking | Relative Importance Score |
|---|---|---|---|
| $A_1$ | 0.14 | 5 | 61% |
| $A_2$ | 0.12 | 6 | 52% |
| $A_3$ | 0.18 | 2 | 78% |
| $A_4$ | 0.23 | 1 | 100% |
| $A_5$ | 0.16 | 4 | 70% |
| $A_6$ | 0.17 | 3 | 74% |

## 7. Sensitivity Analysis

A one-at-a-time sensitivity analysis was conducted to assess the robustness of the provided decisions. In this analysis, the weight of each criterion was systematically varied within the range of 0 to 1, incremented by 0.1, and new scores for the alternatives were calculated. When altering the weight of a single criterion, the weights of the remaining criteria were adjusted proportionally to their original significance. As a result, the total weight of the criteria remained 1 in each scenario.

Figure 4 illustrates the variation in scores of alternatives in response to the changing weights of each criterion within the SFAHP framework. This depiction effectively showcases the sensitivity of each alternative concerning the corresponding criterion.

In the sensitivity analysis depicted in Figure 4a, the interaction between the strategy effectiveness and the weight of the financial criterion is charted. Strategy $A_4$ demonstrated a steady increase in score as the weight of the financial criteria increased, indicating a strong alignment with these criteria. Notably, $A_4$ maintained the highest score throughout, demonstrating its robustness as the favored strategy, regardless of how the financial criterion's importance varied. Strategy $A_3$ started with a relatively high score but slightly decreased as the financial weight increased. This suggests a moderate sensitivity, where $A_3$ somewhat relies on financial criteria but not to a critical extent. The scores for strategies $A_1$ and $A_6$ both showed an upward trend, although they started and ended at different levels. This increase implies that these strategies become more favorable as financial considerations gain emphasis, yet not sufficiently to challenge $A_4$'s dominance. $A_5$'s score decreased over the range, which signifies a sensitivity to the financial criterion that negatively affects its favorability as the financial weight increases. Strategy $A_2$'s score remained relatively stable, with minimal variation. This flat trend indicates a low sensitivity to changes in the financial criterion, suggesting that $A_2$'s evaluation is less dependent on this particular aspect.

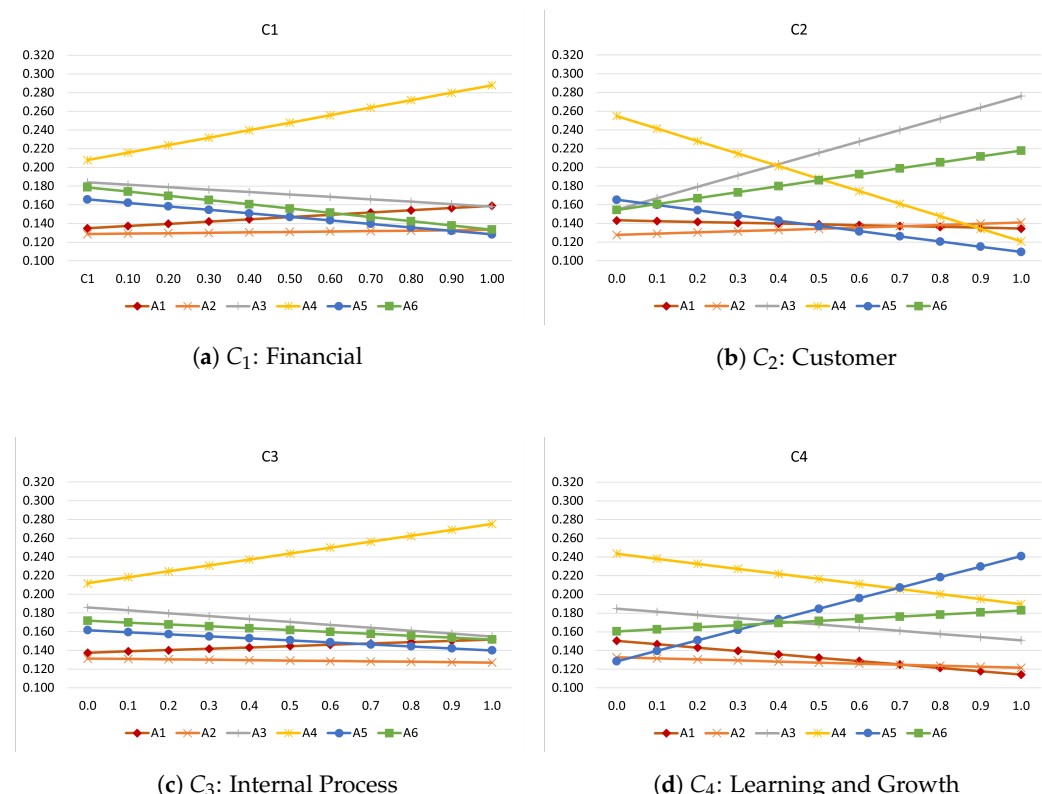

**(a)** $C_1$: Financial

**(b)** $C_2$: Customer

**(c)** $C_3$: Internal Process

**(d)** $C_4$: Learning and Growth

**Figure 4.** Sensitivity analysis for criteria weights.

Figure 4b shows the sensitivity of the strategies to changes in the customer criterion ($C_2$). Strategy $A_4$ began as the leading alternative with the highest score when the weight of the customer criterion was at its lowest, evidencing its superiority with a value of around 0.26. However, as the weight of the customer criteria increased, the score of $A_4$ declined rapidly, plummeting to about 0.12. This descent highlights $A_4$'s negative sensitivity to the customer criterion. However, it maintains its position as the preferred strategy until the customer criterion weight approaches 0.4, illustrating its resilience in scenarios where the customer criterion is less prioritized. Strategy $A_3$, with a starting score of just over 0.15, increased to approximately 0.28, suggesting a positive correlation with the customer criterion. As the weight of this criterion grew, so did $A_3$'s score, indicating its alignment and favorable response to customer-focused considerations. Strategy $A_6$, beginning similarly to $A_3$, witnessed a rise in its score to around 0.22 in response to the increasing weight of the customer criterion. This ascent signifies a positive but less pronounced alignment compared to $A_3$. The score for Strategy $A_5$ displayed a decline from around 0.17 to about 0.11 as the weight of the customer criterion increased. This decline, while not as steep as that of $A_4$, still suggests a considerable negative impact on its favorability due to the customer criterion. Strategies $A_1$ and $A_2$ showed a near-static trend in scores, implying a low sensitivity to the changing weights of customer criterion. Their performance results suggest that these strategies are not significantly influenced by customer considerations, maintaining a steady evaluation irrespective of the emphasis on the customer criterion.

Figure 4c illustrates the response of the strategies to variations in the internal process criterion ($C_3$). Strategy $A_4$ shows the highest performance across all levels, beginning slightly above a score of 0.21 and ascending to nearly 0.28 as the weight of the internal process criterion is maximized. This progression reflects a positive relationship between the strategy's success and the importance placed on internal processes. Strategy $A_1$ starts with a score of around 0.14, and as the weight of the criterion increases, so does its score, reaching up to approximately 0.16. This indicates that $A_1$ performs better with more emphasis on internal processes. On the other hand, Strategy $A_2$'s performance is quite unresponsive to changes in this criterion, beginning and ending around a score of 0.13,

showing a flat trend that suggests a lack of sensitivity to the internal process criterion. Strategies $A_3$, $A_5$, and $A_6$ connect negatively with the internal process criterion. As the weight of this criterion grows, their scores slightly decrease, indicating that these strategies are less effective when internal processes are considered more important. Despite the varying degrees of correlation between the strategies and the internal process criterion, $A_4$ remains the top choice, leading the pack even when its weight is at its lowest. This underlines the robustness of the initial preference for $A_4$, which seems to stand strong against varying emphases on the internal processes.

Figure 4d illustrates how different strategies react to the learning and growth criterion ($C_4$). The score for $A_4$ starts at roughly 0.25 and decreases steadily to near 0.19 as the criterion's weight is fully considered. A trend is observed where the criterion's importance is inversely related to the performance of Strategy $A_4$, reflecting the pattern noted in Figure 4b. Despite this inverse relationship, $A_4$ remains the preferred strategy until the criterion's importance approaches a weight of about 0.7. A remarkable detail from the graph is the sensitivity of the $A_5$ strategy to changes in this particular criterion. While starting at the lowest rank with a score around 0.13 when the criterion weight is 0, $A_5$'s score swiftly climbs as the criterion's importance is elevated. After the weight crosses the threshold of 0.7, $A_5$ takes the lead in the ranking, reaching a score of 0.24 when the criterion weight is at its maximum, positioning it at the top of the priority list. On the other hand, the score for $A_6$ rises moderately from 0.16 to around 0.19 as the criterion weight increases from 0 to 1, indicating a steadier response to the learning and growth criterion compared to other strategies. Meanwhile, $A_1$, $A_2$, and $A_3$ continue to decrease due to their inverse correlation with the criterion. $A_2$'s trajectory is almost horizontal, suggesting it is the least sensitive to learning and growth criterion changes. As the criterion's weight enhances, $A_3$ slips from the second to the fourth rank, impacted by its negative correlation with the criterion and the rising scores of $A_5$ and $A_6$. Despite the negative correlation between $A_4$ and the learning and growth criterion, $A_4$ remains the preferred strategy until the criterion's weight approaches 0.7. However, when the weight of $C_4$ exceeds the 0.7 mark, $A_5$ takes the lead, indicating a notable shift in strategic advantage. Given that typically, no single criterion is expected to dominate with over 70% weight, it can be considered that $A_4$'s top ranking is robust, even in light of its inverse relationship with this criterion.

To summarize, across Figure 4a–d, strategies demonstrate unique reactions to the varying weights of different criteria. In Figure 4a, no strategy substantially challenges $A_4$'s leading position under financial criteria adjustments, confirming the robustness of the initial choice. Figure 4b shows $A_4$'s score declining with increased emphasis on customer criteria, while $A_3$ and $A_6$ improve, and $A_1$ and $A_2$ remain stable, underscoring the need to weigh the customer criteria carefully in strategy selection. In Figure 4c, $A_4$ retains its lead across all levels of criterion weight, reinforcing the original decision's validity. For Figure 4d, $A_4$ upholds its top ranking until the learning and growth criteria weight surpasses 0.7, after which $A_5$ takes the forefront. Given that it is uncommon for a single criterion to dominate so strongly, $A_4$'s resilience suggests significant robustness, even as this criterion's importance escalates.

While the initial summary provides a direct interpretation of our sensitivity analysis outcomes, a deeper understanding emerges when these findings are viewed through the lens of relevant scholarly literature. This broader perspective aligns our analysis with established theories and models in strategic decision making.

For instance, the adaptability and responsiveness observed in our sensitivity analysis, particularly, the varying scores of alternatives like $A_4$ and $A_5$, reflect the insights emphasized by Haktanir et al. [1] on the importance of adaptability in strategic planning within digital transformation contexts. Similarly, the financial criterion's significant impact on strategic effectiveness aligns with the findings of Lee et al. [22], highlighting the critical role of financial aspects in strategic outcomes. The influence of the customer criterion echoes Wang et al.'s [23] insights on customer-centric strategies in the digital era, while the

importance of internal processes and learning and growth criteria reflects the perspectives of Nachtmann et al. [24] and Kaplan and Norton [25], respectively.

In conclusion, the integration of these academic perspectives enriches our understanding of the sensitivity analysis results. It reveals the multifaceted nature of strategic decision making and underscores the need for a flexible and adaptable approach, where strategic effectiveness varies based on the shifting importance of different criteria. This comprehensive view not only validates our strategic choices but also highlights their robustness and adaptability in the face of evolving business environments.

## 8. Conclusions

In this study, which is a key segment of a comprehensive strategic model for adapting to rapidly advancing technology, various planning and decision-making tools, such as the Balanced Scorecard, Objectives and Key Results, SWOT analysis, TOWS, and the Spherical Fuzzy Analytic Hierarchy Process, are utilized together. This research primarily focuses on applying SFAHP and sensitivity analysis for selecting the most suitable strategy among various options, informed by insights derived from SWOT, TOWS, and BSC perspectives.

A significant aspect of this research is utilizing the Spherical Fuzzy Analytic Hierarchy Process (SFAHP). The choice of SFAHP over classical AHP methods was driven by its superior capability to handle the uncertainties and subjective judgments inherent in strategic decision making. This approach is particularly effective in the context of digital transformation, where expert opinions and strategic scenarios often involve a high degree of complexity and ambiguity. The use of SFAHP allowed for a more nuanced assessment of strategies, reflecting real-world scenarios more accurately than traditional methods.

Furthermore, incorporating Spherical Fuzzy Sets in the SFAHP framework marked a significant advancement in handling decision-making ambiguity. This enhanced feature of SFAHP enabled capturing hesitation or uncertainty levels more precisely, contributing to the robustness and reliability of strategic choices. The application of Spherical Fuzzy Sets provided an additional layer of depth in the analysis as evidenced by the sensitivity analysis results. This approach not only validated the selected strategies but also showcased the adaptability and resilience of these strategies in the face of evolving business environments and changing criteria weights.

Digital transformation requires selecting the right strategy, a point strongly emphasized in this research. The application of SFAHP and subsequent sensitivity analysis has deepened our understanding of the chosen strategy's resilience and adaptability. In this context, the analysis of 'Strategy $A_4$' is particularly noteworthy. The way this strategy adapts to changes in criteria weights and the impact of these changes on the strategy's effectiveness provide a concrete example of how the use of SFAHP offers valuable insights into the strategic decision-making process. These findings not only confirm the effectiveness of the selected strategy but also point to the need for adaptability in changing business situations. The comprehensive approach of combining SFAHP with the sensitivity analysis lays a solid groundwork. It helps businesses to navigate the complexities of digital transformation efficiently and assists in ensuring that strategic choices remain relevant and effective amidst technological changes.

The practical application of these tools provides a holistic approach to strategic decision making. It allows for a detailed evaluation of strategies, aligning strategic initiatives with organizational goals. The findings confirm the effectiveness of the selected strategy and underscore the need for a flexible approach to managing digital transformation complexities.

## 9. Limitations and Future Directions

This study focuses on using the Spherical Fuzzy Analytic Hierarchy Process (SFAHP) for selecting digital transformation strategies. It is important to note that relying on expert opinions, as we do, can introduce subjective biases. Also, the study's concentration on specific strategies might limit how widely our findings can be applied to different industries.

Future research should look at applying SFAHP in various industries and developing methods to reduce the subjectivity in expert opinions. It would also be beneficial to explore the long-term effects and adaptability of the chosen strategies under changing business conditions.

This study highlights how SFAHP can be an effective tool in strategic decision making during digital transformation. For managers, the value of using such analytical tools to navigate complexity and uncertainty in strategy selection should be emphasized.

**Author Contributions:** Conceptualization, M.O. and O.D.; Methodology, U.C. and O.D.; Validation, O.D.; Formal analysis, M.O.; Investigation, M.O.; Data curation, M.O. and U.C.; Writing—original draft, M.O., U.C. and O.D.; Supervision, U.C. All authors have read and agreed to the published version of the manuscript.

**Funding:** This research received no external funding.

**Data Availability Statement:** The raw data supporting the conclusions of this article will be made available by the authors on request.

**Conflicts of Interest:** The authors declare no conflicts of interest.

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
