# Peer review of "BSC-Based Digital Transformation Strategy Selection and Sensitivity Analysis"

_mathematics, doi:10.3390/math12020225_

Round 1

Reviewer 1 Report

Comments and Suggestions for Authors

From my point of view, the work is well-done. This paper highlights a key
segment of a comprehensive strategic model developed to address this challenge. The
model integrates lean manufacturing techniques with various planning and
decision-making tools such as the Balanced Scorecard (BSC), Objectives and Key
Results (OKR), SWOT analysis, TOWS, and the Spherical Fuzzy Analytic Hierarchy
Process (SFAHP). Integrating these tools in the proposed model provides businesses
with a well-rounded pathway to manage digital transformation. This paper has a novel
theme, appropriate research methods, clear logic and reasonable structure. However,
the following sections need to be modified:
1. In the introduction, the author should emphasize the innovative discoveries,
research gap and research novelty related to the theme of this paper.
2. In the section 4, the citation of relevant literature should be added to the
method described to increase the authority of the method used. Also, the text
description of Figure 2 should be next to Figure 2, please check the full text.
3. In the section 5, the selection of cases should introduce the reasons for the
selection and the fit with this paper. What is the basis for the three key dimensions
divided? It should be supported by relevant literature. In addition, what is the basis of
the four elements analyzed by using the SWOT analysis method?
4. Figure. 3 can be optimized as appropriate. Also, check whether the font size is
consistent in the Figure 3, such as the "C13 Return on investment" font size should be
consistent with others.
5. In the section 7, The discussion of the results of sensitivity analysis should be
combined with relevant literature, rather than a simple explanation of the phenomenon,
and the deep causes should be explored.
6. In Conclusion section, The limitations and shortcomings of the research
should be supplemented, and the outlook of future research should be proposed. And a
part of the “Management enlightenment” should be added to analyze the practical
significance of the conclusion of this paper.
7. The results and conclusions of the paper should be related to the existing
literature. And, please check the reference format, such as “Haktanir, E.; Kahraman,
C.; Seker, S.; Dogan, O. Future of Digital Transformation. In Intelligent Systems in
Digital Transformation: Theory and Applications; Springer, 2022; pp. 611–638.”
should be modified as “Haktanir, E.; Kahraman, C.; Seker, S.; Dogan, O. Future of
Digital Transformation. In Intelligent Systems in Digital Transformation: Theory and
Applications; Springer, 2022; pp. 611–638.”

Author Response

We attached a detailed response file. Thank you for your valuable comments on improving our study.

Reviewer 2 Report

Comments and Suggestions for Authors

A comprehensive strategic model is developed to help companies to navigate new technological trends. Lean manufacturing techniques with various planning and decision-making tools are involved in the proposed model, such as the Balanced Scorecard, Objectives and Key Results, SWOT analysis, TOWS, and the Spherical Fuzzy Analytic Hierarchy Process. The inclusion of these tools in the proposed model provides organizations with a thorough strategy for steering and supervising digital transformation. To enhance the paper, the authors need to make the following revisions:

1. Why use complex spherical fuzzy sets as the form of information representation? The motivation needs to account for the applicability and necessity of using it. The same problem exists in the choice of other basic methods, so the motivation should be restated.

2. The contribution needs to be elucidated, encompassing what issues the proposed method can address or what deficiencies in existing methods it can rectify.

3. How can it be improved if consistency is not met? Consistency determines whether the ranking results have good logic, but the relevant narration is lack in this paper. There are some recent developments over the consistency of AHP. The authors could reference the following papers to enhance their discussions, see Managing transitivity and consistency of preferences in AHP group decision making based on minimum modifications. Information Fusion, 2021, 67: 125-135. Analytic hierarchy process rank reversals: causes and solutions. Annals of Operations Research, 2023, doi:

4. According to the results of Table 8, some additional insights and underlying reasons should be discussed, rather than simply elucidating the contents of the table. The ordinate title of Figure 4 needs to be added.

5. What are the advantages of the proposal over existing studies?

Author Response

(The authors gave the same response as above.)

Round 2

Reviewer 2 Report

Comments and Suggestions for Authors The revisions basically cleared my concerns, and I am fine with the current version.